# Docking Design of the Different Microcapsules in Aqueous Solution and Its Quantitative On-Off Study

**DOI:** 10.3390/polym15051131

**Published:** 2023-02-24

**Authors:** Hongfei Tan, Dan Zhao, Mingxing Liu, Zongguo Hong, Jingxue Liu, Kang Dai, Xincai Xiao

**Affiliations:** 1School of Pharmacy, South-Central Minzu University, Wuhan 430074, China; 2National Demonstration Center for Experimental Ethnopharmacology Education, South-Central Minzu University, Wuhan 430074, China; 3School of Food and Biological Engineering, Hubei University of Technology, Wuhan 430068, China; 4The College of Art and Science, The Ohio State University, Columbus, OH 43210, USA

**Keywords:** docking systems, dual carrier, temperature response, intermolecular hydrogen bonds, phase transitions

## Abstract

To avoid risk, spacecraft docking technologies can transport batches of different astronauts or cargoes to a space station. Before now, spacecraft-docking multicarrier/multidrug delivery systems have not been reported on. Herein, inspired by spacecraft docking technology, a novel system including two different docking units, one made of polyamide (PAAM) and on of polyacrylic acid (PAAC), grafted respectively onto polyethersulfone (PES) microcapsules, is designed, based on intermolecular hydrogen bonds in aqueous solution. VB12 and vancomycin hydrochloride were chosen as the release drugs. The release results show that the docking system is perfect, and has a good responsiveness to temperature when the grafting ratio of PES-g-PAAM and PES-g-PAAC is close to 1:1. Below 25 °C, this system exhibited an “off” effect because the polymer chains on the microcapsule’s surface produced intermolecular hydrogen bonds. Above 25 °C, when the hydrogen bonds were broken, the microcapsules separated from each other, and the system exhibited an “on” state. The results provide valuable guidance for improving the feasibility of multicarrier/multidrug delivery systems.

## 1. Introduction

Stimulus-responsive drug delivery systems have become popular because of their controlled drug release, ability to improve the bioavailability of drugs, and reduced toxicity [1]. Systems based on various external stimuli (such as light [2,3], temperature [4], pH [5] and electricity [6]) have been studied. It is well known that signals from changes in body temperature can be easily captured. Therefore, many drug delivery systems are specifically designed to produce the rapid release of a drug in response to an increase in body temperature caused by a disease (e.g., inflammation or cancer) in a positive feedback mode, releasing the drug rapidly when the external temperature increases and slowing the release of the drug when the temperature decreases. Meanwhile, with further research on some diseases, single-drug treatment modalities are no longer able to satisfy clinical drug delivery needs. Single-drug approaches may not achieve the best efficacy for some diseases, and may produce large toxic side effects. In contrast, multidrug therapy can reduce the toxicity of drugs, exploit the synergistic effect of drugs and enhance the therapeutic effect, thus delaying and reducing the generation of drug resistance [7]. As examples of multidrug therapy, a combination of antibiotics is used to treat Helicobacter pylori infections [8] and a combination of chemotherapeutic agents [9,10] is used for cancer treatment. Therefore, combining the advantages of stimulus-responsive systems and multidrug therapy has been the focus of ongoing research.

The multidrug delivery systems reported thus far allow the simultaneous loading of multiple drugs, and are generally based on the partitioning of a single carrier lumen and loading of different drugs into their corresponding closed compartments, such as layer-by-layer structures [11,12], double- or multi-chamber structure [13,14,15], and polymeric micelles [16,17]. However, many multi-chamber systems will not be able to load multiple hydrophilic or lipophilic drug molecules simultaneously due to the hydrophobic interface, which will limit the choice of drugs for combination therapy. Meanwhile, the preparation process of these multi-compartmental drug delivery systems is complicated; the continuous separation of the components of these multi-chamber systems is not yet possible, thus limiting the flexibility in drug delivery and mass production. Besides this, these complex multi-chamber structures are more prone to defects during the preparation process, which can lead to multiple drug molecules interacting in the carrier and thus reducing the drug activity.

Spacecraft docking technology can avoid relevant risks by transporting batches of different astronauts or cargoes to the space station. Space stations generally include core modules, cargo spacecraft, manned spacecraft, and other modules for experiments and other functions. The core module can dock with various laboratory modules, manned spacecraft, and cargo spacecraft. The spacecraft is docked to a space station to transport personnel and supplies to the space station. The spacecraft can be used multiple times, and a damaged spacecraft can be replaced by another spacecraft, thereby increasing the utilization of the rest of the station. 

Herein, inspired by the space-docking mechanism, with its structure and advantages, for the first time, a novel docking model system in aqueous solution, in which different substances (acting as drugs) are loaded into different microcapsules, was explored. The different microcapsules could interact with each other so that drug release is prevented by default, and the dual drugs are released simultaneously when the microcapsules are separated (Figure 1). This dual drug delivery system overcomes the irreversibility and inflexibility of traditional multidrug delivery systems, while overcoming the challenge of not releasing multiple drug simultaneously due to the hydrophilic interface. Once one carrier is damaged during transport or storage, and it can be replaced with a new unit of the same microcapsules, similar to the maintenance of a machine. Different drugs can also be replaced for combination therapy at any time, making clinical drug administration more convenient.

To accomplish docking among different microcapsules, the docking force is key. In this work, hydrogen bonds were chosen as the docking force. Polyamide (PAAM) and polyacrylic acid (PAAC) have also been chosen for their ability and reversibility to provide the hydrogen bonds. These are well-known polymeric materials with thermal responsiveness [18,19]. At a temperature lower than the upper critical solution temperature (UCST) [20], PAAC forms intermolecular hydrogen bonds with PAAM [21]. However, at a temperature higher than the UCST, the intermolecular hydrogen bonds break, and the polymer chains dissociate [22]. Most previously reported PAAC–PAAM-type drug delivery systems are based on an interpenetrating polymer network (IPN) [23,24,25,26,27]. However, for the first time, previous studies have qualitatively [28] and quantitatively [29] explored the coupling effect of PAAC/PAAM with a non-IPN structure. The non-IPN system exhibited the same behaviors of intermolecular hydrogen bonds as the IPN system, and could achieve rapid drug release (narrower phase temperature ranges). Therefore, in the present study, we also used PAAM and PAAC, and grafted PAAC or PAAM chains onto polyethersulfone (PES) microcapsules with a finger–pore–channel structure, designated as PES-g-PAAC or PES-g-PAAM, respectively. The finger–pore–channel structure can avoid delayed the drug release, which occurs with the labyrinth-like structure of the microcapsules shells [30]. At a temperature lower than the UCST, the intermolecular hydrogen bonds between PAAC and PAAM polymer chains in PES-g-PAAC and PES-g-PAAM microcapsules, respectively, hinder the release of drug molecules, resulting in an “off” effect. At a temperature higher than the UCST, the hydrogen bonds between the PAAC and PAAM polymer chains break, the two microcapsules separate from each other, and the drug molecules can cross the surface of the PES microcapsules easily, resulting in an “on” state. Drug release experiments are used to verify the responsiveness of a drug delivery system to temperature, and to explore the quantitative relationship between the grafting rates of the microcapsules. The results of this study provide some guidance for the study of a multidrug/multicarrer delivery system.

## 2. Materials and Methods

### 2.1. Materials

Polyethersulfone (PES, purity 99%, MV40000) was provided by Jida High-tech Materials Co., Ltd. (Changchun, China). Vitamin B12 (VB12, 1 g) was provided by Saiguo Biotechnology Co., Ltd., Guangzhou, China. Vancomycin hydrochloride (0.1 g) was provided by Shanghai Yuanye Bio-Technology Co., Shanghai, China. Other chemical reagents, such as Potassium bromide (KBr, purity 99%), N,N-Dimethylacetamide (DMAC, purity 99%), polyethylene glycol 400 (PEG400, purity 99%), methanol (CH_3_OH, purity 99%), concentrated sulfuric acid (H_2_SO_4_, purity 99%), polyvinylpyrrolidone (PVP, K30), formic acid (HCOOH, purity 99%), Ceric ammonium nitrate (CAN, purity 99%), acrylic acid (AAC, purity 99%), acrylamide (AAM, purity 99%), and *N,N*-Dimethylbisacrylamide (BIS, purity 99%) were all provided by Sinopharm Chemical Reagent Co., Ltd. (Shanghai, China). Reverse osmosis-derived water and deionized water (D.I. water) were used in all experiments.

### 2.2. Preparation of the PES Microcapsules

The finger pore-shaped PES microcapsules were prepared by the “sol-gel” phase conversion method [31]. Briefly, a 100 mL beaker was taken and 3.69 g of PES was dissolved in 25 mL of DMAC under stirring, then 0.275 g of LiCl, 5.63 g of PEG400 and 6 g of PVP were added sequentially and stirred until complete dissolution. The solution was incubated at 3 °C for a period of time until the bubbles in the solution disappeared. Then, the solution was cured by dropping it into water through a syringe at a curing temperature of 30 °C. It was kept at 30 °C and cured for 30 min to obtain finger–hole PES microcapsules.

### 2.3. Surface Corrosion of the PES Microcapsules

For the easy grafting of PAAC/PAAM onto the surface of PES microcapsules, the surfaces of the prepared PES microcapsules were etched with DMAC by soaking the PES microcapsules in DMAC for 40 s, removed and washed in kerosene several times, and then placed in water to cure. Then, the PES microcapsules were soaked in a water bath at 37 °C for more than 4 days and then washed with water 2–3 times a day until the water became clear. Finally, the surface-etched PES microcapsules were dried in an oven at 40 °C and set aside.

### 2.4. Surface Grafting of PAAM/PAAC on the PES Microcapsules

The prepared and processed PES microcapsules (0.1000 g) were soaked in 20% formic acid solution for 24 h. To a three-neck flask, 150 mL of distilled water, cerium nitrate amine, H_2_SO_4_, and *N*,*N*’-methylenebisacrylamide were added. The initiation system was composed of ceric amine nitrate and H_2_SO_4_, and BIS was used as the crosslinking agent. After peroxidation, nitrogen gas was passed, and soaked AAC/AAM and PES microcapsules were added 5 min later. The reaction mixture was stirred at 100 rpm in the nitrogen environment for 24 h in a water bath at 60 °C until the solution became clear.

### 2.5. Calculation of the Grafting Rate of the PES Microcapsules

The grafted PES microcapsules were soaked in pure water for 48 h (the water was changed 2–3 times a day) and dried at 40 °C until the mass remained unchanged. The grafting rate was calculated according to the following formula:(1)$$G\%=\frac{m_{g}-m_{0}}{m_{0}}\times 100\%$$
where *G*% denotes the grafting ratio; *m*_0_ and *m_g_* represent the mass of PES microcapsules before and after grafting, respectively.

### 2.6. Characterization of the Microcapsules

The samples used for FTIR spectroscopy were prepared with KBr pellets, the microcapsules were mixed and ground with KBr for compression, and the spectra were recorded with ANTARIS II (THERMO). SEM analysis of the platinum-coated surface (model SU8010, Japan) was used to compare the surface and cross-section of the microcapsules before and after grafting. The microcapsules before and after grafting were subjected to corrosion in DMAC solution for 24 h. The morphological structures before and after corrosion were compared.

### 2.7. The Standard Curves of VB12 and Vancomycin Hydrochloride 

To determine the release process accurately, the model drugs (VB12 and vancomycin hydrochloride) had to be calibrated. In total, 0.01 g of VB12 or vancomycin hydrochloride was weighed in a beaker and dissolved with a small amount of deionized water. The dissolving VB12 solution or vancomycin hydrochloride solution were added into a 100 mL volumetric flask and diluted to the scale line to obtain the two mother liquors at a concentration of 100 µg·mL^−1^. In total, 1 mL of VB12 mother liquor or vancomycin hydrochloride mother liquor was measured, and we diluted them 2.5-, 3-, 3.5-, 4-, 4.5- and 5-fold by adding ultrapure water. The absorbance of different concentrations of VB12 was measured by a UV spectrophotometer at 361 nm, and the absorbance–concentration standard curve of VB12 was plotted. The absorbance of different concentrations of vancomycin hydrochloride was measured at 281 nm, and then the absorbance–concentration standard curve of vancomycin hydrochloride was plotted.

### 2.8. Single Drug Release Experiments

To explore the switching effect of the system and the quantitative relationship, VB12 was chosen as the model drug. The grafting rates of PES-g-PAAC and PES-g-PAAM microcapsules were calculated separately, and the PES-g-PAAM microcapsules with different grafting rates and PES-g-PAAC microcapsules with different grafting rates were combined to obtain eight sets of microcapsule combinations with different grafting rates. For each group, five replicates of PES-g-PAAC microcapsules and five replicates of PES-g-PAAM microcapsules were soaked in 20 mL of VB12 solution (50 µg·mL^−1^) for 24 h. Then, 20 mL of distilled water was added at 5 °C, 10 °C, 15 °C, 20 °C, 25 °C, and 30 °C. The absorbance of aliquots of the solutions was determined at different time intervals; the aliquot was poured back after measurement. The concentration of VB12 was calculated from the measured absorbance, and a concentration–time graph was plotted.

### 2.9. Dual Drugs Release Experiments

To prove the switching effect of the system comprising dual drugs, VB12 and vancomycin hydrochloride were chosen as model drugs. The two groups of PES-g-PAAC (10%, 15%, 20% and 24%) and two groups of PES-g-PAAM (12% and 17%) were combined. For each group, five replicates of PES-g-PAAM microcapsules were soaked in 20 mL of VB12 solution (50 µg·mL^−1^) for 24 h. Five replicates of PES-g-PAAC microcapsules were soaked in 20 mL of vancomycin hydrochloride solution (50 µg·mL^−1^) for 24 h. After soaking, the two were mixed at 5 °C, 10 °C, 15 °C, 20 °C, 25 °C, and 30 °C. The absorbance of aliquots of the solutions was determined at different time intervals; the aliquot was poured back after measurement. The concentrations of VB12 and vancomycin hydrochloride were calculated separately from the measured absorbance, and concentration–time graphs were plotted.

## 3. Results 

### 3.1. Characterization of the PES Microcapsules

PES microcapsules with finger–pore shapes were prepared by the phase transition method. Then, PAAC or PAAM chains were grafted on the surface of PES microcapsules by the chemical grafting method. Figure 2A displays the FT-IR spectra of different types of PES microcapsules. Carbonyl stretching resonances at 1670 cm^−1^ and 1716 cm^−1^ were observed for PAAM and PAAC pure systems, respectively. Two distinct absorption bands were observed compared to the blank PES microcapsules. PES-g-PAAC showed a significant νC=O stretching vibration at 1716 cm^−1^, which was due to the νC=O absorption in the carboxyl group shifting to lower waves due to the intermolecular hydrogen bonding. In contrast, PES-g-PAAM showed a distinct νC=O stretching vibration at 1670 cm^−1^ (amide I band). This also proves that different polymers were grafted on the PES microcapsules.

Figure 2B shows the scanning electron microscopy (SEM) images of different types of PES microcapsules. The surfaces of the unmodified PES microcapsules were relatively smooth, tight, and neat, and the cross-section shows tight straight open pores and a thick surface, which would not favor graft reactions and could affect drug release. The surfaces of the PES microcapsules dissolved with *N*,*N*-dimethylacetamide (DMAC) were relatively rough and uneven, with many holes. The straight pores were maintained, but the surface was thin, indicating that corrosion occurred. The surfaces of both PES-g-PAAC microcapsules and PES-g-PAAM microcapsules were attached to an area of porous, uneven, irregular surface with similar straight pores, which were very different from other surfaces, indicating that a polymer shell was grafted on the surface of the corroded PES capsules.

Blank PES microcapsules dissolved immediately in DMAC after 1 min, whereas the other grafted PES microcapsules became transparent and maintained their spherical shape even after 24 h. After the complete dissolution of PES in the grafted microcapsules at 24 h, transparent spheres of PAAC or PAAM remained (Figure 2C), indicating that in DMAC solution, the PES microcapsules were dissolved by the DMAC solution, while the PAAC and PAAM polymers grafted on the PES microcapsules could not be dissolved, resulting in the formation of transparent polymer spheres. When combining Fourier transform infrared (FTIR) spectroscopy and SEM, the PAAC and PAAM were successfully grafted onto the surfaces of the PES microcapsules.

### 3.2. Single Drug Release In Vitro

To explore the switching effect and quantitative relationship of this drug delivery system, we combined PES-g-PAAC microcapsules with different grafting rates and PES-g-PAAM, and selected VB12 as a model drug. VB12 was loaded into two different microcapsules and drug release experiments were performed at different temperatures (5 °C, 10 °C, 15 °C, 20 °C, 25 °C and 30 °C). The cumulative drug concentrations in groups A1, A2, A5, A6 and A7 were lower when the temperature was lower than 25 °C, and significantly higher when the temperature was higher than 25 °C (Figure 3). This was because the PES-g-PAAC microcapsules and PES-g-PAAM microcapsules contacted each other during the drug release process. When the temperature was lower than the UCST, the grafted PAAC and PAAM produced hydrogen bonds at the contact sites of the two different microcapsules, which underwent a “zipper effect” at the contact sites and prevented the release of VB12 from the microcapsules. When the solution temperature was close to or higher than the UCST, the hydrogen bonds at the contact sites of PES-g-PAAC microcapsules and PES-g-PAAM microcapsules broke, the interacting parts separated, and the drug molecules could thus easily cross the surface of the PES microcapsules, leading to an increased drug concentration.

In contrast, there was no such regularity for the other groups in Figure 3. The other groups did not show a regular relationship between cumulative drug release and temperature. To investigate the reasons for this phenomenon, we counted the grafting rates of all groups; the results show that the drug release was greatest when the grafting ratio of PES-g-PAAM microcapsules to PES-g-PAAC microcapsules was 0.68–1.25 (close to 1:1), with the most pronounced hydrogen bond switching effect of PAAM and PAAC, which was the same as the ratio cited in the literature [22,23,24,25,26,27] for the IPN structure. This was because, when the contact areas of the two microcapsules were the same and the amount of -CONH_2_ on the PES microcapsules was closer to the amount of -COOH, the intermolecular hydrogen bonding at the contact site of the microcapsules and the hydrogen bonding switch were optimal. However, when the grafting rates of PES-g-PAAC and PES-g-PAAM differed too much, incomplete hydrogen bonding at the contact site or a lack of hydrogen bonding did not produce a switching effect. The ratio in our experiment with VB12 was very close to 1:1, but not exactly 1:1, probably because the PES microcapsules were not prepared in a completely homogeneous state and the grafting rates of PAAM or PAAC were not exactly the same.

### 3.3. Dual Drugs Release In Vitro

The above drug release experiments reveal that when a drug was loaded in this dual carrier drug delivery system, a switching effect was produced at different temperatures, and by a statistical analysis of the grafting rates of two different microcapsules, it was found that the switching effect of drug release was more obvious when the grafting rates of the two microcapsules were relatively close to each other. Therefore, we sought to further test this hypothesis and to explore the switching effect when loaded with two different drugs. In the experiments, VB12 and vancomycin hydrochloride, both water-soluble drug molecules, were chosen as model drugs; these do not react with each other and have non-interfering maximum UV wavelengths for measuring the cumulative drug solubility. We loaded VB12 and vancomycin hydrochloride into PES-g-PAAC microcapsules and PES-g-PAAM microcapsules with different grafting rates, respectively. Drug release experiments were performed at different temperatures (5 °C, 10 °C, 15 °C, 20 °C, 25 °C and 30 °C). 

From the results of the dual drug release experiments, we can observe that the combination of PES-g-PAAM with 12% grafting rate and PES-g-PAAC with 10% grafting rate, and the combination of PES-g-PAAM with 12% grafting rate and PES-g-PAAC with 15% grafting rate, also showed a significant switching effect. When these two microcapsules with close grafting rates were combined, the microcapsules loaded with VB12 and vancomycin, respectively, both showed larger cumulative drug concentrations at temperatures higher than the UCST. This also indicates that the greater cumulative concentration of the drug at temperatures above the UCST is not generated by one carrier alone, but by the hydrogen bonding at the contact sites of both PES-g-PAAM and PES-g-PAAC microcapsules to control the drug release from both microcapsules. Of course, we can also clearly observe that when the grafting rate of the PES-g-PAAC microcapsules differs significantly from that of PES-g-PAAM microcapsules, the cumulative release concentrations of VB12 and vancomycin do not show such regularity with respect to temperature (Figure 4).

We also found such a trend in groups C2 and C3, where the grafting rates of PES-g-PAAM and PES-g-PAAC were also relatively similar, and the cumulative release rates of both VB12 and vancomycin hydrochloride increased at temperatures higher than the UCST. In contrast, the release of both drugs did not show a clear regularity with temperature in the C1 and C4 groups, where the grafting rates differed considerably (Figure 5).

From the statistics and analysis of the above two groups of dual-drug release experiments, it can be found that when the grafting ratios of PES-g-PAAM and PES-g-PAAC were 0.85–1.2 (again, close to 1:1), they also showed some positive feedback temperature responsiveness, i.e., the release of both drugs from the dual carriers was accelerated when the temperature was higher than the UCST, while the release of both drugs was blocked when the temperature was lower than the UCST. The temperature responsiveness was less pronounced when the grafting rates of PES-g-PAAM-loaded drugs and PES-g-PAAC-loaded drugs were more different, which also indicates that the grafting rate of this dual-carrier drug delivery system will determine whether a switching effect can be produced. The release by the dual drug system also suggests that at close grafting rates, the hydrogen bonds generated at the contact sites of the two microcapsules break when the external temperature is higher than the UCST, allowing the release of different drugs within both capsules, rather than the effect produced by a particular microcapsule.

## 4. Conclusions

In conclusion, inspired by spacecraft docking, we successfully explored a novel docking-type drug-release system that contained two separate drugs, loaded in two separate microcapsules, and the two microcapsules released the drugs only when they were separated from each other. The drug release profiles at different temperatures were studied using VB12 and vancomycin hydrochloride as model drugs, loaded into PES microcapsules grafted with PAAC and PAAM, respectively. The release results display that the docking system is perfect and has good responsiveness to temperature. When the ratio of grafted PAAM to PAAC on the surface of PES microcapsules was close to 1:1 and the external temperature was lower than the UCST of PAAC-PAAM, the intermolecular hydrogen bonds formed between PES-g-PAAC and PES-g-PAAM inhibited the release of drug molecules. However, when the temperature was higher than the UCST, the hydrogen bonds broke, and the drug molecules could be easily released, meaning the drug delivery system was in the “on” state. With the evolution of microcapsule materials, further research on the proposed system will lead to new methods and ideas for achieving clinical combination therapies.

## Figures and Tables

**Figure 1 polymers-15-01131-f001:**
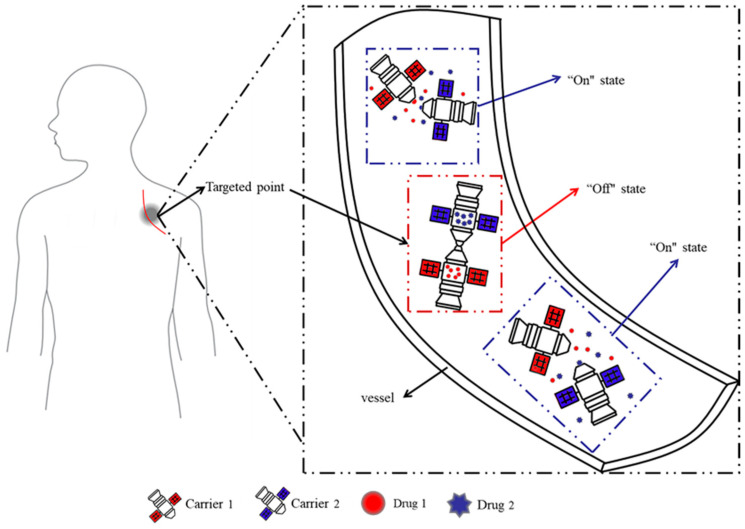
Design ideas of the proposed docking system.

**Figure 2 polymers-15-01131-f002:**
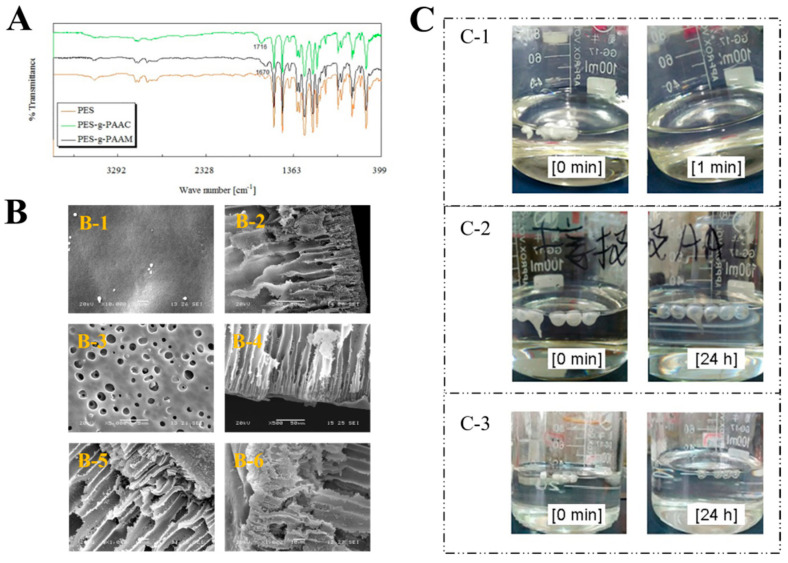
Characterization of the microcapsules. (**A**) The FT−IR spectra of blank PES microcapsules, PES microcapsules grafted with PAAC, and PES microcapsules grafted with PAAM. (**B**) Scanning electron microscopy images of ungrafted and grafted microcapsules. (**B-1**) Surfaces of PES microcapsules (Scale bar 1 μm). (**B-2**) Cross-sections of the PES microcapsules (Scale bar 50 μm). (**B-3**) Surfaces of modified PES microcapsules (Scale bar 5 μm). (**B-4**) Cross-sections of modified PES microcapsules (Scale bar 50 μm). (**B-5**) Cross sections of PES-g-PAAC microcapsules (Scale bar 10 μm). (**B-6**) Cross-sections of PES-g-PAAM microcapsules (Scale bar 10 μm). (**C**) The images of different PES microcapsules after being immersed in DMAC. (**C-1**) Blank PES microcapsules. (**C-2**) PES-g-PAAC microcapsules. (**C-3**) PES-g-PAAM microcapsules.

**Figure 3 polymers-15-01131-f003:**
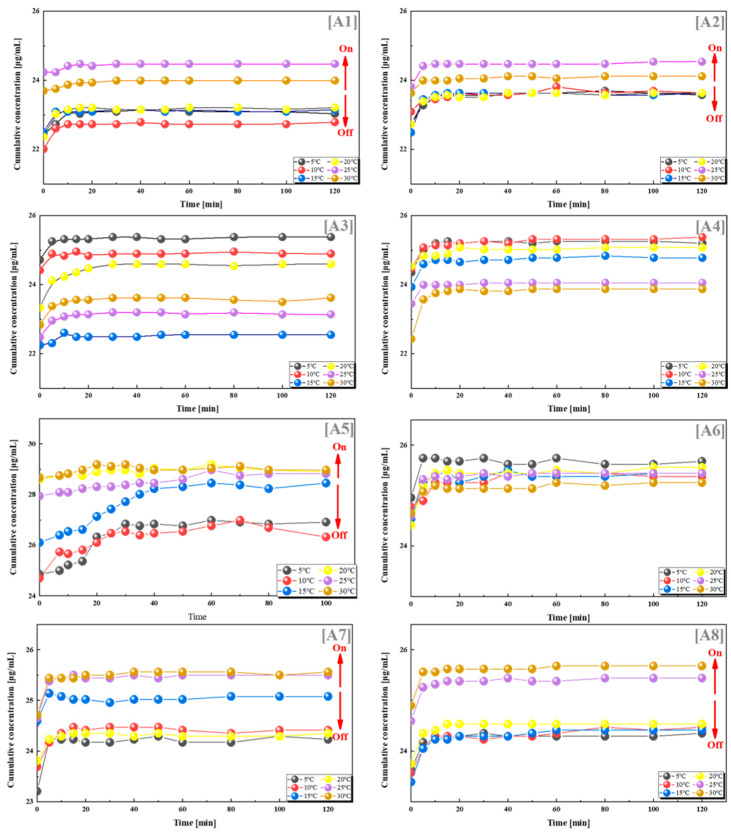
VB12 release from systems with different combinations of grafting rates at different temperatures. (**A1**) 12–9%; (**A2**) 12–15%; (**A3**) 12–21%; (**A4**) 12–25%; (**A5**) 13–14%; (**A6**) 17–9%; (**A7**) 17–21%; (**A8**) 17–25%. All the above combinations are for PES-g-PAAM microcapsules and PES-g-PAAC microcapsules with different grafting rates.

**Figure 4 polymers-15-01131-f004:**
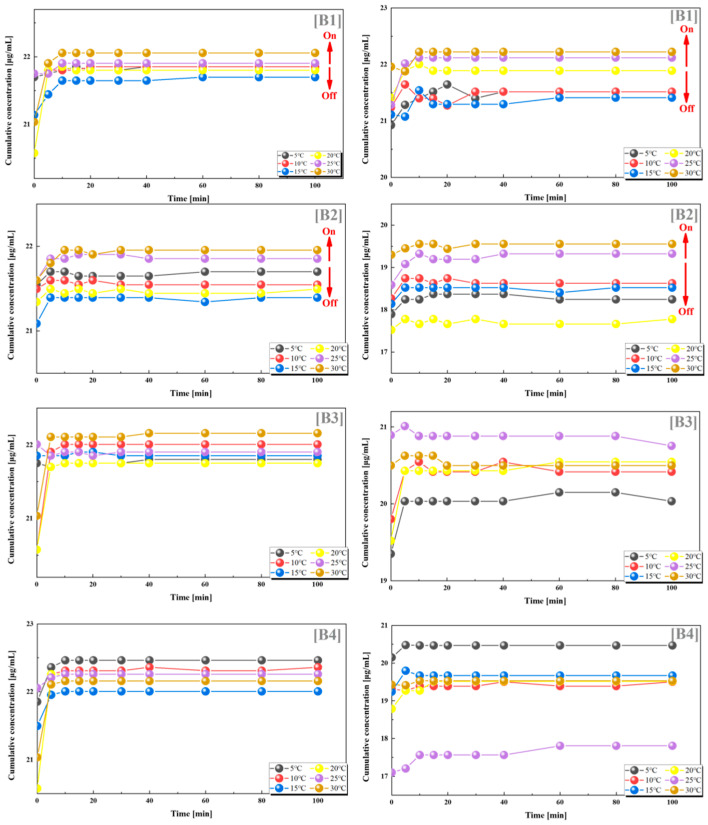
Release of VB12 (left) and vancomycin hydrochloride (right) from systems with different combinations of grafting rates at different temperatures. (**B1**) 12–10%; (**B2**) 12–15%; (**B3**) 12–20%; (**B4**) 12–24%. All the above combinations are for PES-g-PAAM microcapsules and PES-g-PAAC microcapsules with different grafting rates.

**Figure 5 polymers-15-01131-f005:**
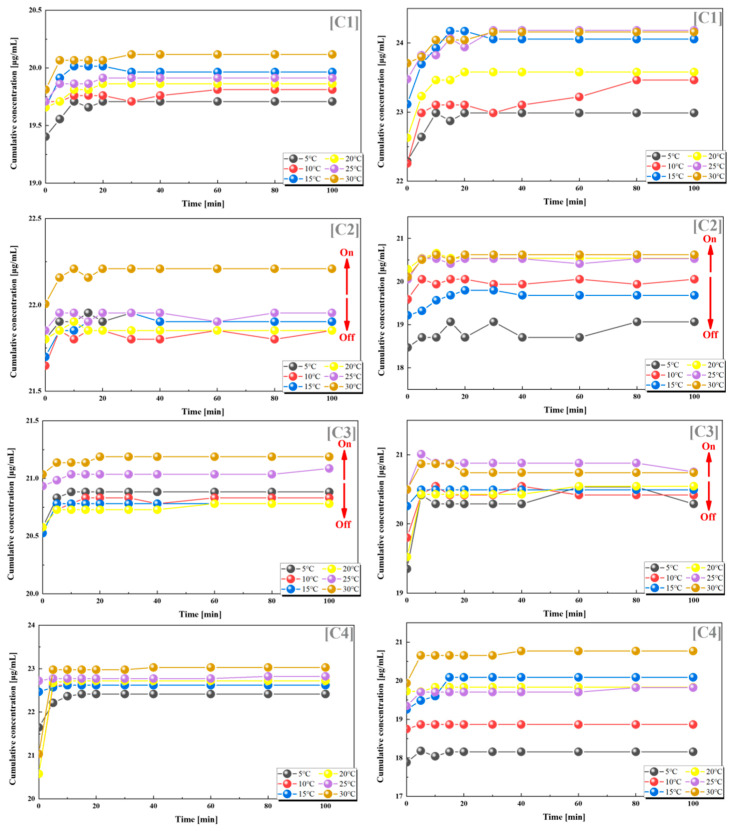
Release of VB12 (left) and vancomycin hydrochloride (right) from systems with different combinations of grafting rates at different temperatures. (**C1**) 17–10%; (**C2**) 17–15%; (**C3**) 17–20%; (**C4**) 17–24%. All the above combinations are for PES-g-PAAM microcapsules and PES-g-PAAC microcapsules with different grafting rates.

## Data Availability

Data presented in this study are available on request from the corresponding author.

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
