# Peer review of "Docking Design of the Different Microcapsules in Aqueous Solution and Its Quantitative On-Off Study"

_polymers, 2023, doi:10.3390/polym15051131_

Round 1
Reviewer 1 Report
The study “Docking design of the different microcapsules in aqueous solution and its quantitative on-off study” form Tan et al. explores an interesting topic. But I order in order to increase the usefulness and significance of the study, it needs a revision before being considered suitable for readers and there are some points to overcome for acceptance.
Please provide high quality images in each figure or graphical representation (figure 1 and 3).
Use consistent style in each figure representation. Figure numbering is mismatched. Figure 2 is missing.
In section 2.3, the authors said that PES was washed with kerosene. A scientific explanation should be given for this statement.
Page 6, line 236, The ratio in the experiment with VB12 was very close to 1:1 but not exactly 1:1, the author should give the exact ratio with the proper explanation.
Why did the group with a large difference in grafting rate not show a clear regularity in the release of both drugs with respect to temperature?
The reason why groups C2 and C3 have relatively close grafting rates, including their responsiveness to the temperature?
Conclusion is very shallow; it should contain more outcomes of the current research work.
Double check the way of adding references in the main text body and reference section as per journal guidelines. Minor typo mistakes in reference section of the manuscript. Need to be check and correct carefully.
Reviewer 2 Report
This manuscript can be accepted for publication after the authors provide sufficient responses to the following comments:
1. What is the size of the pore?
2. For the drug release, kinetic should be correlated using several well-known drug release models.
Round 2
Reviewer 1 Report
Author addressed all comments carefully so I endorse this article for the publication.
Reviewer 2 Report
The authors have provided sufficient responses to my comments, therefore it can be accepted for publication.